# Scale-Variant Flight Planning for the Creation of 3D Geovisualization and Augmented Reality Maps of Geosites: The Case of Voulgaris Gorge, Lesvos, Greece

**Ermioni-Eirini Papadopoulou** [1,*] , **Apostolos Papakonstantinou** [2] , **Nikolaos Zouros** [1] **and Nikolaos Soulakellis** [1]

[1] Department of Geography, University of the Aegean, 81100 Mytilene, Greece; nzour@aegean.gr (N.Z.); nsoul@aegean.gr (N.S.)
[2] Department of Marine Sciences, University of the Aegean, 81100 Mytilene, Greece; apapak@aegean.gr
* Correspondence: epapa@geo.aegean.gr; Tel.: +30-22510-36876

**Abstract:** The purpose of this paper was to study the influence of cartographic scale and flight design on data acquisition using unmanned aerial systems (UASs) to create augmented reality 3D geovisualization of geosites. The relationship between geographical and cartographic scales, the spatial resolution of UAS-acquired images, along with their relationship with the produced 3D models of geosites, were investigated. Additionally, the lighting of the produced 3D models was examined as a key visual variable in the 3D space. Furthermore, the adaptation of the 360° panoramas as environmental lighting parameters was considered. The geosite selected as a case study was the gorge of the river Voulgaris in the western part of the island of Lesvos, which is located in the northeastern part of the Aegean Sea in Greece. The methodology applied consisted of four pillars: (i) scale-variant flight planning, (ii) data acquisition, (iii) data processing, (iv) AR, 3D geovisualization. Based on the geographic and cartographic scales, the flight design calculates the most appropriate flight parameters (height, speed, and image overlaps) to achieve the desired spatial resolution (3 cm) capable of illustrating all the scale-variant details of the geosite when mapped in 3D. High-resolution oblique aerial images and 360° panoramic aerial images were acquired using scale-variant flight plans. The data were processed using image processing algorithms to produce 3D models and create mosaic panoramas. The 3D geovisualization of the geosite selected was created using the textured 3D model produced from the aerial images. The panoramic images were converted to high-dynamic-range image (HDRI) panoramas and used as a background to the 3D model. The geovisualization was transferred and displayed in the virtual space where the panoramas were used as a light source, thus enlightening the model. Data acquisition and flight planning were crucial scale-variant steps in the 3D geovisualization. These two processes comprised the most important factors in 3D geovisualization creation embedded in the virtual space as they designated the geometry of the 3D model. The use of panoramas as the illumination parameter of an outdoor 3D scene of a geosite contributed significantly to its photorealistic performance into the 3D augmented reality and virtual space.

**Keywords:** UAS; scales; flight planning; 3D mapping; 3D geovisualization; AR; geosites; panorama

## 1. Introduction

Cartography is the science, technology, and art of studying and creating maps. All types of drawings, diagrams, and 3D models that depict the earth at any scale are considered drawings [1]. New technologies that emerged in the 20th century, such as geoinformatics, remote sensing, 3D modeling, multimedia, and the internet, came to change the cartographic process and to reshape the definition of cartography [2]. Cartographic visualizations act as an interdisciplinary tool as they can represent information and data from various scientific fields [3]. In addition to the new technological tools that provide digital geographic data and spatial information to cartography and GIS science, another

important contribution to the above was the science of information visualization [4]. Information visualization acted synergistically with the above sciences, creating the field of geovisualization [5]. Geovisualization is defined as the visualization of geospatial data [6], and more specifically the use of visual representations to interpret spatial patterns and relationships in complex data [7]. Geovisualizations can be achieved through spatial-related tools and algorithms when applied to data with spatial, temporal, and semantic value [6]. Geovisualization gave a new impetus to cartography by incorporating the part of interaction into it [8]. Where technological means, such as multimedia, analytical and computational methods, and web mapping, were used to achieve the interaction.

Multimedia refers to the combination of media and information with an interactive form of connection. It uses different media to interactively convey information, such as text, sound, graphics, animation, and video [9]. Thus, new tools and means are needed to communicate the information and transfer knowledge efficiently to the audience when illustrating spatio–temporal phenomena. Essential elements that serve both needs are computers and mobile devices, which are nowadays accompanied by the internet and fast mobile networks. In addition, the integration of "multimedia technologies" with GIS allow the provision of more realistic 3D representations of the temporal dimension of spatial objects and phenomena [10]. The adaptation of multimedia technology in cartographic products provides a different way of displaying and better understanding geographic phenomena. The means used to achieve multimedia cartography change and evolve over time [11]. Therefore, cartographic creation and design have adapted the new visualization tools in order to improve a user's spatial information communication and interaction [7,12].

One of the factors contributing significantly to cartography science in recent years was the development of new geospatial data-collection tools [13–15]. One of these tools was the unmanned aerial system (UAS), which transformed the collection process of high-resolution images in a time-efficient, highly repetitive, and safe way over hard-to-reach and dangerous areas [16–22]. Several recent publications have described methods and techniques for the 3D mapping of spatio–temporal phenomena using UAVs [20,23–31]. Currently, UAVs are a viable option for high-resolution, remote-sensing data collection, applicable in a wide range of scientific, agricultural, and environmental fields [18,29,32–38]. UASs can bridge the gap from local observations to regional mapping using earth-observation (EO) data [39]. UASs can provide high-quality information as well as time and cost-efficient monitoring of spatio–temporal phenomena at a local scale [40–42]. Along with UASs as data-collection tools, data processing algorithms have also evolved. Algorithms such as structure-from-motion and multi-view stereo (MVS) were developed [43,44], helping in automatically processing a large number of high-resolution UAS images to create 3D information [45–48].

During the few last years, the 3D geovisualization of geographic information created using the above processes and algorithms has been realized through augmented and virtual reality depictions. Augmented reality (AR) was defined by Azuma in 1997 [49] as the augmentation of real space by introducing additional information through an artificial medium, such as mobile platforms. Virtual reality is defined as a computer-generated simulation of a three-dimensional world that can be experienced in a believable manner by people wearing special electronic equipment such as headsets with integrated displays or gloves with sensors etc. [50,51]. In recent years, more and more geographical applications of geolandscaping and analysis use these specific techniques in different fields of application [29,52–55]. In geosite promotion and management there are references to augmented and virtual reality applications where users can browse through virtual space and observe specific geological formations remotely [56]. Furthermore, in other cases, users can visit a geosite in situ and receive 3D or 2D information for the area through an application installed on their mobile phone [20,57–62]. However, few studies exist in the literature that investigate how the quality of the 3D model of outdoor geological formations affects the data collection, considering the geographical and cartographic scale of the geosite and issues of optical variables on the 3D virtual space.

This paper investigates the scale issues related to data acquisition and the scale-variant flight parameters that affect the creation of 3D geovisualizations used in AR maps. The relationship between the geographic and cartographic scale, the data spatial resolution, and the produced 3D models of the geosites are studied in more depth. In addition, the lighting of the 3D models is examined as an optical variable in the 3D space as well as the contribution of the 360° panoramas as environmental lights.

This study's contribution lies in the creation of scale-variant data acquisition flight plans for mapping and depicting geosites in AR. Additionally, it illustrates flight plans for 3D mapping using 360° panoramas and aerial images for the 3D geovisualization of complex geomorphological structures with intense lighting differentiation. More specifically, the use of panoramas as the key illumination parameter of an outdoor 3D scene of geosite is proposed as it contributes significantly to its photorealistic performance in the 3D virtual space. This ensures the visual improvement of the original 3D models in the AR environment for the 3D geovisualization and mapping of areas with geological interest.

## 2. Study Area

The Voulgaris Gorge geosite is located in the western part of Lesvos island in the North Aegean Region, Greece. The study area was selected due to its impressive geomorphology, it is an awe-inspiring part of the Voulgaris Gorge that covers an area of five hectares (Figure 1). The Voulgaris river gorge is a typical example of an erosional geosite. It was formed on the Lower Miocene volcanic rocks of the Vatousa volcano by the interaction of two sets of geological faults, a NW–SE trending fault zone and the younger E–W trending normal faults [52,53]. The Voulgaris river water flow carves the fragmented volcanic rock, creating a deep ravine with steep rocky sides and constant erosion. In addition, the Voulgaris Gorge's geosite integrity is threatened by the construction of the new Kalloni–Sigri highway. The geosite may soon receive interventions due to the impending widening of the adjacent road. The Voulgaris river gorge geosite requires high-resolution mapping to capture its natural beauty and integrity.

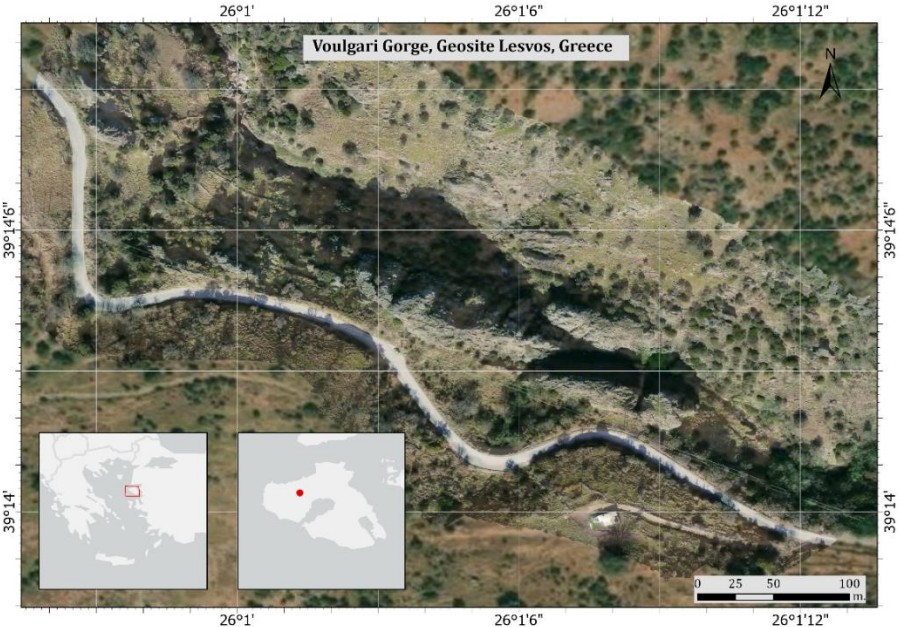

**Figure 1.** Map of the study area: geosite of Voulgaris Gorge on Lesvos Island, Northeastern Aegean region, Greece.

## 3. Methodology

The methodology applied in this project consisted of four pillars: (i) flight planning, (ii) data acquisition, (iii) data processing, (iv) 3D geovisualization (Figure 2). In the first pil-

lar scale-variant flight plans and flight parameters were designed and selected accordingly. In the second, the collection of high-resolution images using a UAS from the field was performed. The aerial images collected were processed using computer vision algorithms to create 3D point clouds, 3D models, DSMs, orthomosaics, and panoramas of the study area. Finally, the 3D models and the panoramas were combined in a 3D environment for the 3D photorealistic geovisualization of the geosite in AR.

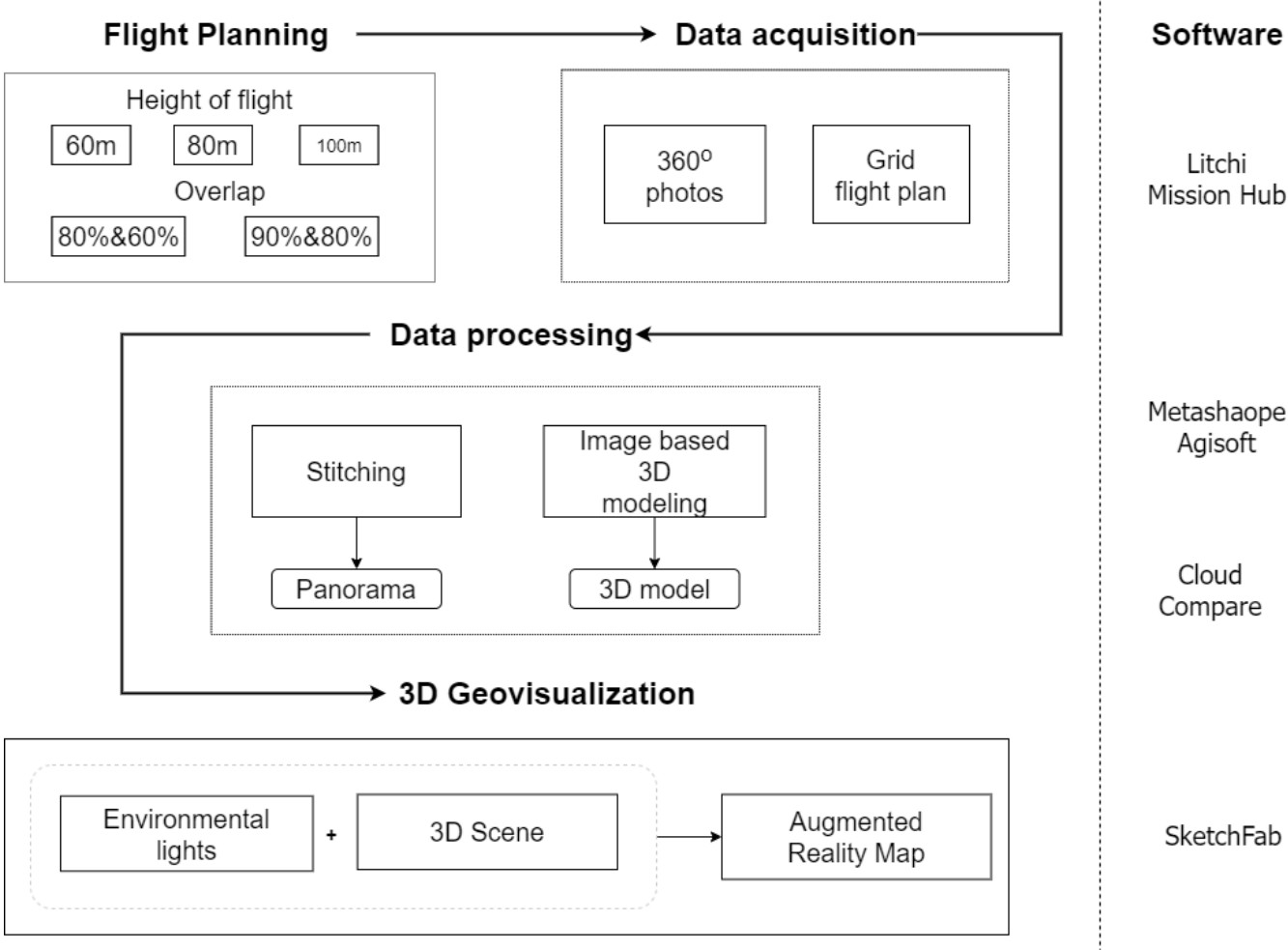

**Figure 2.** Methodology flow chart: flight planning, data acquisition, data processing, 3D geovisualization and the software that was used.

### 3.1. Cartographic Scale-Variant Flight Planning

A map is a symbolic representation of specific characteristics of an area depicted on a flat surface and includes qualitative and quantitative information about the world in a simple, visual way. To understand maps properly specific parameters need to be understood thoroughly, with the map scale being the most important parameter of them. In this context, the most critical step to address in the 3D mapping of a geosite is defining the visualization's appropriate cartographic scale, which directly affects the UAS flight planning. Furthermore, the geographical area to be mapped should be defined as it affects the selection of the drone, endurance capabilities, and sensor characteristics.

The required ground sampling distance (GSD) relies on the map's cartographic scale on which the 3D geovisualization will be placed as an AR item. Thus, the sizes of the features of interest are correlated by map production processes that are necessarily approximate. These processes result in a variety of error effects that apply to geometric details on a map.

The cartographic scale (sc) of derivative maps is based on the flight height (h) and focal length (fl) in aerial data acquisition and is calculated using Equation (1) [63].

$$sc = fl/h \tag{1}$$

According to Tobler, "The rule is: divide the denominator of the map scale by 1,000 to obtain the detectable size in meters. The resolution is one-half of this amount" [64]. This makes the spatial resolution the most critical parameter when mapping. The spatial resolution of the produced DSM and orthomosaics from the UAS data processing is inseparably linked to the GSD of the images. Thus, they are directly affected by the flight altitude and image characteristics (resolution, sensor size, and focal length). As GSD is driven by the spatial scale of the AR 3D geovisualization, the flight parameters and the image selection and settings require trade-offs for survey optimization in order to achieve the best parameterization. The GSD of the images is calculated using Equation (2) [65].

$$GSD = (sw*h)/(fl*iw) \tag{2}$$

where GSD is ground sampling distance (m), sw is sensor width (mm), h is the flight height (m), fl is the focal length (mm), and iw is image width (pixels).

For this study, considering that the cartographic product's selected scale was 1:400, the acceptable spatial error was calculated to be 10 cm. Furthermore, based on the human eye's visual acuity that is ¼ of a millimeter, the smaller spatial entity visually detectable in a map with a scale of 1:400 is 10 cm. In this case, the acceptable spatial error for all raster derivatives of aerial data (DSM and orthomosaic) is GSD/3 [66] thus 3.33 cm. Using this value, the maximum flight altitude was calculated using Equation (2) at 100 m. When designing the scale-variant flights three heights were selected using 100 m as an altitude threshold, which gives images that need a spatial resolution of 3 cm. The three selected heights were 60 m, 80 m, and 100 m. The first flight altitude chosen was 100 m as it was the height that produced the smallest amount of aerial images that had the desirable GSD. The 60 m flight altitude selected was the value at which the curve began to flatten, meaning that was the value where the drone could acquire the minimum number of images to cover the selected area with the lowest spatial resolution (1.8 cm). Moreover, the 80 m was finally selected as the third flight altitude as an intermediate height between 60 m and 100 m.

Another significant parameter in flight planning is the determination of the forward and side image overlap, particularly for structure-from-motion (SfM) photogrammetric reconstructions, which require features observed in multiple images for developing DSMs, orthomosaics, and 3D models. More specifically, two different overlap percentages were analyzed: (i) 90% and 70%, and (ii) 80% and 60%, for forward and side overlap, respectively.

In this study, the correlation between the number of images and the flight altitude was examined based on the GSD. The graph in Figure 3 depicts this correlation. The curve started to flatten after a certain flight altitude, meaning that from 60 m and above (at a higher flight altitude), for the selected study area of five hectares using 80% frontal and 60% side lap, the number of images acquired does not change significantly.

The scale-variant flight scenarios that were examined are presented in Table 1. Scenario 1 was at 60 m altitude resulting in images with GSD of 1.8 cm and 80% and 60% overlap. Scenario 2 was a flight from 60 m with 90% and 70% overlap and GSD 1.8 cm. Scenario 3 and Scenario 4 had an 80 m flight altitude, the GSD was 2.4 cm, and the forward and side overlap was 80% and 60% and 90% and 70%, respectively. The fifth scenario (Scenario 5) was the 100 m altitude (80% and 60%) with GSD 3cm, and the sixth scenario (Scenario 6) was at the same altitude with 90% and 70% overlap.

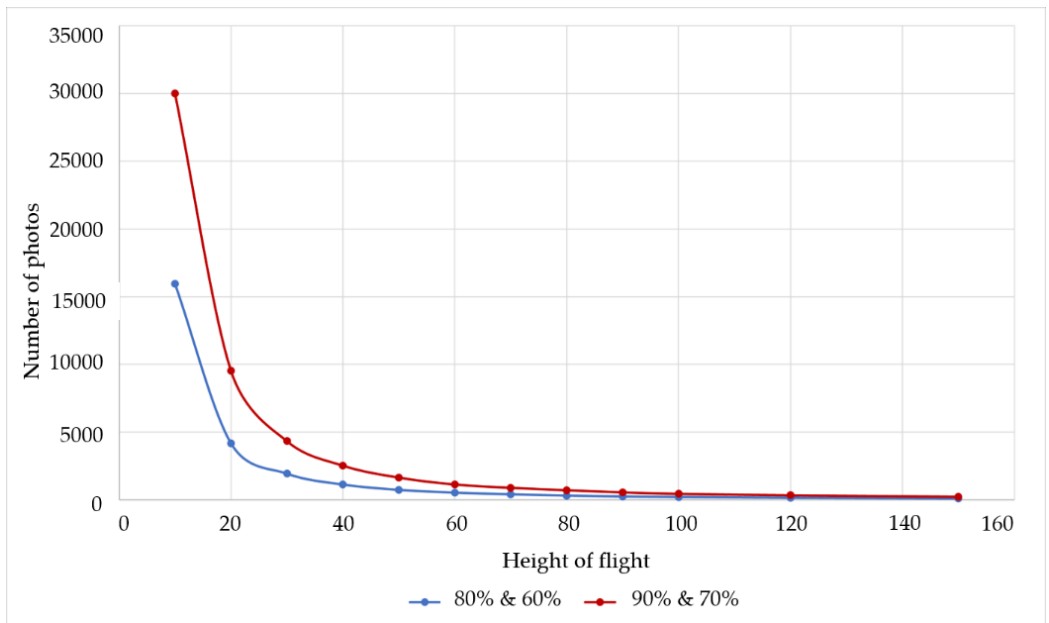

**Figure 3.** Graph showing flight altitude (x-axis) and the number of images (y-axis): relationship chart for the flight planning with an overlap rate of 80% forward and 60% sideways.

**Table 1.** The six scenarios' characteristics of height of flight, overlap, and GSD.

|  | Height of Flight | Overlap | GSD |
|---|---|---|---|
| Scenario 1 | 60 m | 80% and 60% | 1.8 cm |
| Scenario 2 | 60 m | 90% and 70% | 1.8 cm |
| Scenario 3 | 80 m | 80% and 60% | 2.4 cm |
| Scenario 4 | 80 m | 90% and 70% | 2.4 cm |
| Scenario 5 | 100 m | 80% and 60% | 3 cm |
| Scenario 6 | 100 m | 90% and 70% | 3 cm |

*3.2. Data Acquisition*

The aerial survey was performed using a DJI Phantom 4 Pro quadcopter equipped with a mechanical shutter, 20-megapixel camera attached to a 3-axis gimbal to maintain the camera at a predefined level. The camera sensor had a 24 mm focal length (35 mm format equivalent) and a one-inch complementary metal oxide semiconductor (CMOS) sensor. The UAS was using an inertial measurement unit (IMU), a barometer, combined with a GPS/GLONASS positioning system achieving hover accuracy of ±0.5 m vertically and ±1.5 m horizontally. Finally, the UAS was powered using an intelligent flight battery that provided approximately 23 min of flight time under normal conditions.

For each of the above scenarios, a different flight plan was designed using the Litchi Mission Hub flight planner [67] that met all the parameters (flight altitude and overlap) that emerged from the investigation into the scale issues. Initially, two flight planning parameters were investigated: the flight altitude and the overlap of the images. Overall, six flight plans were created at three different flight altitudes of 60 m, 80 m, and 100 m, and in two different overlaps, 80% front overlap with 60% side overlap and 90% front overlap with 70% side overlap (Figure 4), for the 3D mapping of the gorge. The design pattern of all flights was in grid form and the aircraft's orientation was to the next waypoint. The aircraft camera was vertical to the ground (nadir), and the flights maintained a constant altitude above the ground. The front overlap of the images was adjusted by the aircraft's speed with the assumption that the UAS would capture images every 2 sec. The characteristics of the flights performed for data collection are described in Table 2. More specifically, both Flight

1 and 2 were realized at 60m height with different front and lateral overlaps; thus, 282 and 462 images were collected, respectively. Both Flight 3 and 4 were executed at 80 m having overlap values of 80% and 60% and 90% and 70%, respectively. The number of images collected from these flights were 307 and 391, respectively. As for the flights realized at 100 m altitude, 201 and 275 images were acquired having the same overlap values as at lower altitude.

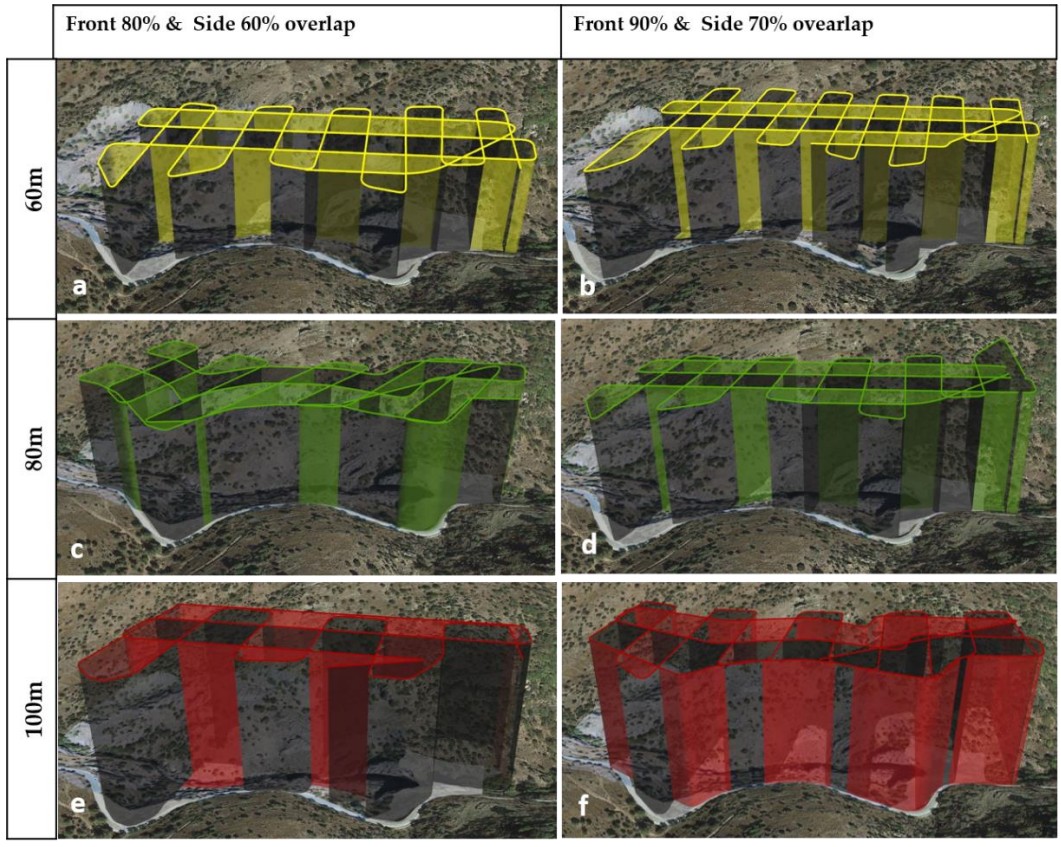

**Figure 4.** Flight plans for data collection: (**a**) the 60 m flight with 80% front overlap and 60% side overlap, (**b**) the 60 m flight with 90% and 70% overlap, (**c**) the 80m flight with 80% and 60% overlap, (**d**) the 80 m flight with 90% and 70% overlap, (**e**) the 100 m flight with 80% and 60% overlap, and (**f**) the 100 m flight with 90% and 70% overlap.

**Table 2.** The number of images collected depending on the flight altitude and the overlap percentage.

| | Height of Flight | Overlap | Duration | Number of Images | Flight Speed |
|---|---|---|---|---|---|
| Flight 1/scenario 1 | 60 m | 80% and 60% | 12 min | 307 images | 20 km/h |
| Flight 2/scenario 2 | 60 m | 90% and 70% | 20 min | 462 images | 12 km/h |
| Flight 3/scenario 3 | 80 m | 80% and 60% | 10.5 min | 282 images | 21 km/h |
| Flight 4/scenario 4 | 80 m | 90% and 70% | 18 min | 391 images | 14 km/h |
| Flight 5/scenario 5 | 100 m | 80% and 60% | 8 min | 201 images | 25 km/h |
| Flight 6/scenario 6 | 100 m | 90% and 70% | 12 min | 275 images | 22 km/h |

The data's geographical location was determined using the Phantom 4 Pro's GPS, as the exact geographical location was not required for the 3D geovisualization of the results in the virtual space. Moreover, in all acquired images, GPS coordinates were embedded in the EXIF metadata. We defined that when the 3D model was rendered through the virtual space, it was transferred to an arbitrary coordinate system.

In addition, images were collected using the panorama mode from Litchi [67], with which horizontal, vertical, and spherical panoramas were captured. A total of three panoramas were captured at a flight height of 60 m, 80 m, and 100 m, and the images were arranged in eight columns per 45° and five rows per 24°, with each panorama consisting of 43 images. Restrictions were observed during the data collection process, one of which was the sun's angle as the orientation of the study area was north, which resulted in certain parts not being adequately lit. Another factor that influenced and intensified this specific phenomenon was the gorge's intense geometry, resulting in strongly shaded parts of the geosite which created negative slopes.

### 3.3. Data Processing

Quality control of the images was first executed on a visual level using the image quality index (IQI). Images that were shaken or blurred were excluded from the image-based 3D modeling process. The selected images were then aligned using the structure-from-motion algorithm, creating a sparse point cloud. Then the dense point cloud was created by applying the multi-view stereo algorithm. The triangulation irregular network (TIN) method was then applied to the dense point cloud through which the points were joined together to create a 3D digital object in the mesh form (Figure 5). Then, the pixels of the high-resolution images collected by the UAS to generate the orthomosaic were orthorectified. The processing of data were the same in all data sets, and the same settings were used.

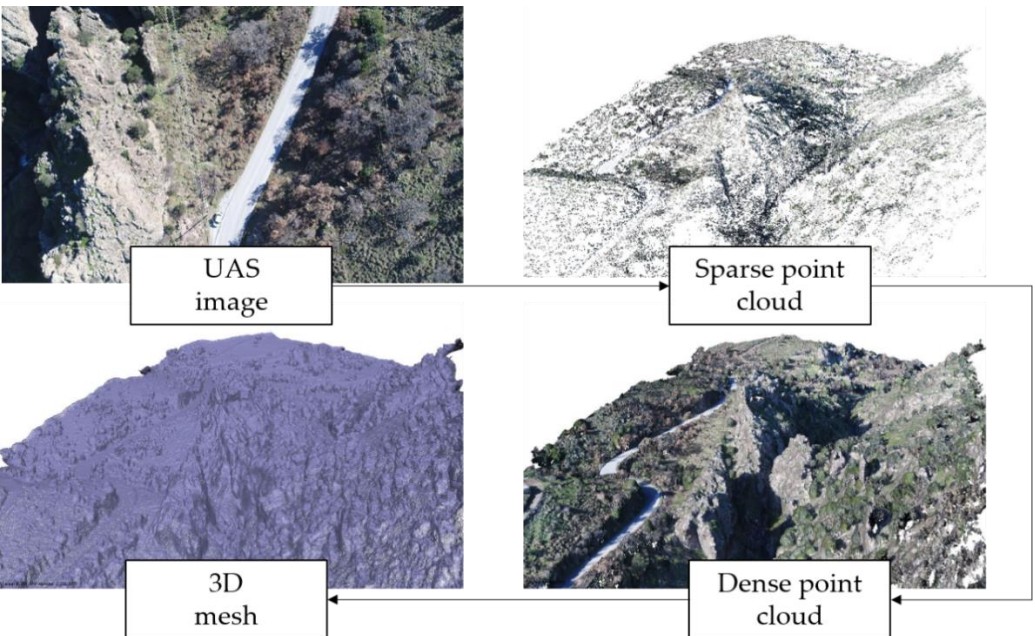

**Figure 5.** Data processing using SfM and MVS.

The 3D digital object was then rendered and wrapped with a photorealistic texture from the high-resolution images collected during the data acquisition. The next stage in data processing was the creation of the 360° panorama. Initially, a camera group was created to add all the photos captured from the Litchi mode panorama [67]. The group of photos was then defined as a camera station and the alignment of the images was performed. The photos needed to overlap to perform this specific process (alignment) to create a complete panorama. The editing process proceeded by stitching the spherical panoramas generated from multiple images taken from the same camera position. Figure 6 shows the set of panoramic images (60 m) stitched according to the spherical model and the camera's angle used to create the panorama. The panorama borders were adjusted so that there were no empty sections at the top and bottom of the staple while the sun's

photograph was placed in the central position of the panorama. The software used for data processing was Agisoft's Metashape [68].

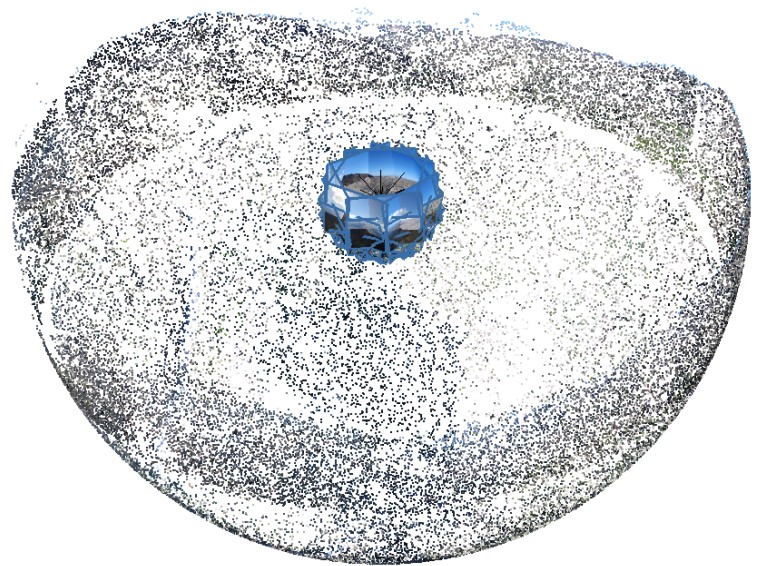

**Figure 6.** Spherical panoramas generated from images (60 m).

### 3.4. 3D Modeling Results

The results obtained from the above process were six 3D point clouds, six textured 3D models, three panorama images, and the corresponding orthomosaics. More specifically, as shown in Table 3, the datasets created from a flight altitude of 60 m had the higher resolution (1.91 cm) and the largest number of points and faces in the dense point cloud and 3D mesh as expected. The GSD for the scenarios realized at 80 m and 100 m flight altitudes were 2.53 and 2.98 cm, respectively. As both had a coarser resolution, the values of points and faces were smaller than those of scenario 1 and 4. As expected, when the flight altitude got higher, the number of points and faces of the dense point cloud and 3D mesh got lower due to the coarser information depicted in the drone images (Figure 7).

In addition, three different panoramas were created, one from each flight altitude (60 m, 80 m, 100 m). The first panorama was created from 43 images collected at the 60 m altitude and consisted of 23,073 × 11,537 pixels (Figure 8). The second (80 m) and third (100 m) panoramas also consisted of 43 images with the same amount of pixels.

**Table 3.** Number of points in the dense point cloud and mesh faces for each data set.

| Dataset/Scenario | Dense Point Cloud | 3D Mesh | Orthomosaic |
|---|---|---|---|
| Overlap 80% and 60% | | | |
| Scenario 1 (60 m) | 25,945,545 points | 5,189,002 faces | 1.91 cm |
| Scenario 3 (80 m) | 21,989,749 points | 4,397,949 faces | 2.53 cm |
| Scenario 5 (100 m) | 19,294,674 points | 3,858,934 faces | 2.98 cm |
| Overlap 90% and 70% | | | |
| Scenario 2 (60 m) | 28,702,725 points | 5,740,520 faces | 1.91 cm |
| Scenario 4 (80 m) | 25,003,619 points | 5,000,723 faces | 2.53 cm |
| Scenario 6 (100 m) | 19,768,631 points | 3,953,726 faces | 2.98 cm |

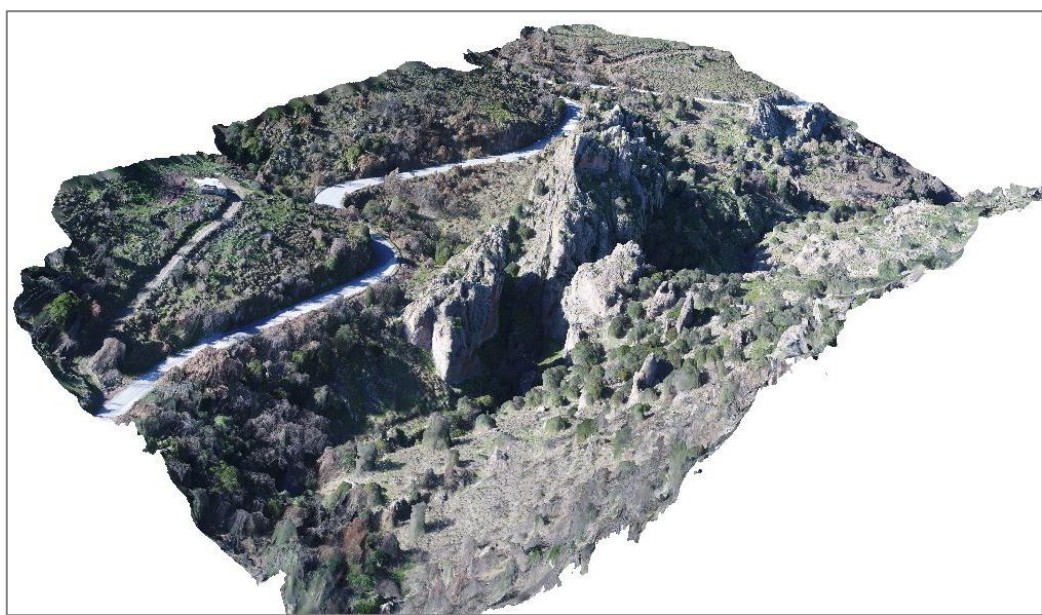

**Figure 7.** 3D model of the Voulgaris gorge, from scenario 4 (80 m altitude and 90% and 70% overlap).

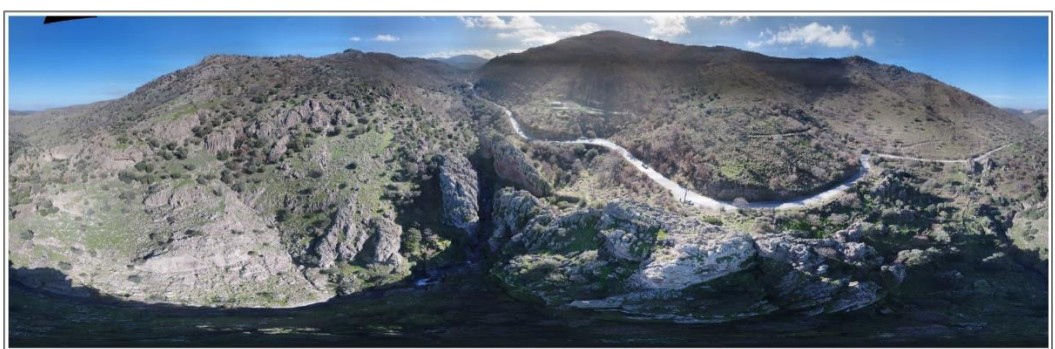

**Figure 8.** Panorama of images from a 360° shot at 80 m.

Evaluation of 3D Models

The selection of the 3D model and the panorama for the geovisualization were based on the most detailed and geometrically correct 3D mapping of the geosite. As shown in Figure 9, gaps appeared in the point cloud on the vertical and dark sides in a gorge section with irregular slopes. The point cloud created from the images collected from the 60 m flight with 80% front and 60% side overlap showed significant gaps on its north side. The same was observed in all the point clouds (80 m and 100 m) created from the acquired images with the same overlap (80% and 60%). The significant gaps in the 3D point clouds significantly affected the creation of 3D models as they determined their geometry. The point clouds constructed from the flights' images with 90% front and 70% side overlap showed a greater number of points on the vertical sides at the respective flight altitudes. However, in Figure 9b,f discontinuities were observed in the point cloud, which indicated the absence of information in these specific sections. In the case of Figure 9d, the created point cloud did not show significant gaps and discontinuities in the distribution of points on the steep slopes and represented the geometry of the geosite more precisely.

The visual observation made through the quality control mentioned above was confirmed by quantitative means of the cloud-to-cloud (C2C) distance algorithm of cloud comparison [69]. Initially, all (six) 3D point clouds were introduced into the Cloud Compare software (CloudCompare 2.11, 2020) and were segmented to the same extent. The point clouds were then registered on a reference cloud. The point cloud with the highest density was selected as the reference cloud and used to register all the other clouds. Then the C2C

distance algorithm was applied, defining the cloud with the largest number of points as a basis for comparison and a fixed distance at which the two clouds' points were calculated. As a reference cloud, the 3D point cloud created from the images collected at 60 m flight height with 90% and 70% overlap and side overlap was selected (scenario 2), and a radius of r = 0.5 m was applied (Figure 10b). Figure 10 illustrates the results of the C2C comparison.

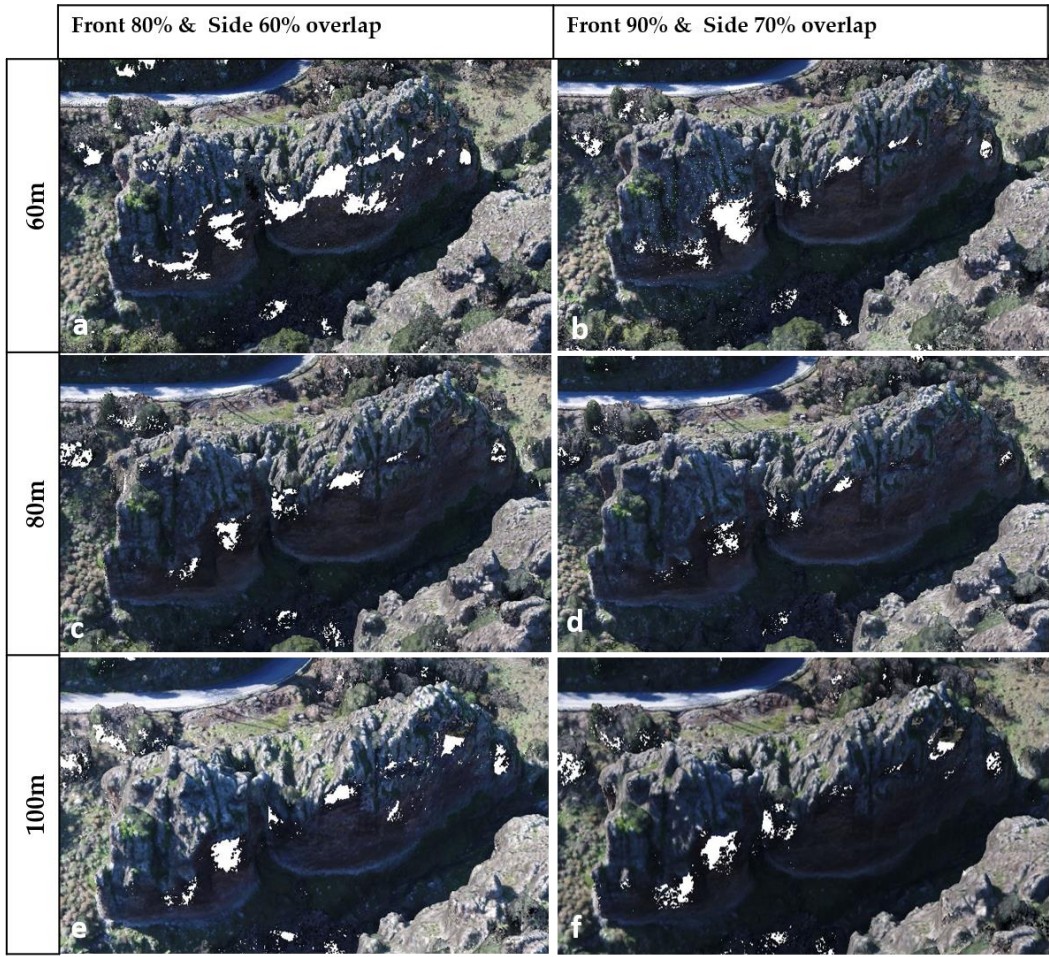

**Figure 9.** 3D dense point cloud: (**a**) from the flight of 60 m with 80% front overlap and 60% side overlap, (**b**) from the flight of 60 m with 90% and 70% overlap, (**c**) from the flight of 80 m with a 80% and 60% overlap, (**d**) from the 80 m flight with 90% and 70% overlap, (**e**) from the 100 m flight with 80% and 60% overlap, and (**f**) from the 100 m flight with 90% and 70% overlap.

Figure 10a depicts the differences between the clouds created from the data acquired from the altitude of 60 m and with an overlap rate of 80% and 60% and 90% and 70%, respectively. The absence of points in certain parts of the gorge denotes the lack of information on the 3D model. Green points that appear in the western and upper part indicate points having a distance of 20 cm. Furthermore the existence of numerous blue points designates that the two point clouds were similar in the biggest part of their extent (Figure 10a).

From the comparison of the reference point cloud (scenario 2) with the scenario 3 point cloud (Figure 10c) we can assume that the center part of the gorge has no differences as a plethora of blue points having distances less than 5 cm exist in this part. Additionally, in the upper center part of the scenario 3 point cloud, the red parts depict the differences of the two point clouds with distances higher than 45 cm.

The comparison of the point clouds in scenario 2 with those of scenario 4 had the following results (Figure 10d): (a) they had the smallest number of holes thus the geometry of the gorge was mapped better in scenario 4, and (b) in the west upper part of the point cloud a red part existed denoting that the two point clouds had significant differences

(greater than 45 cm). These red parts may also be translated as the lack of points (holes) in the scenario 2 point cloud, and (c) that the center part of the gorge had no differences as a plethora of blue points having distances between 0 and 10 cm existed in this part.

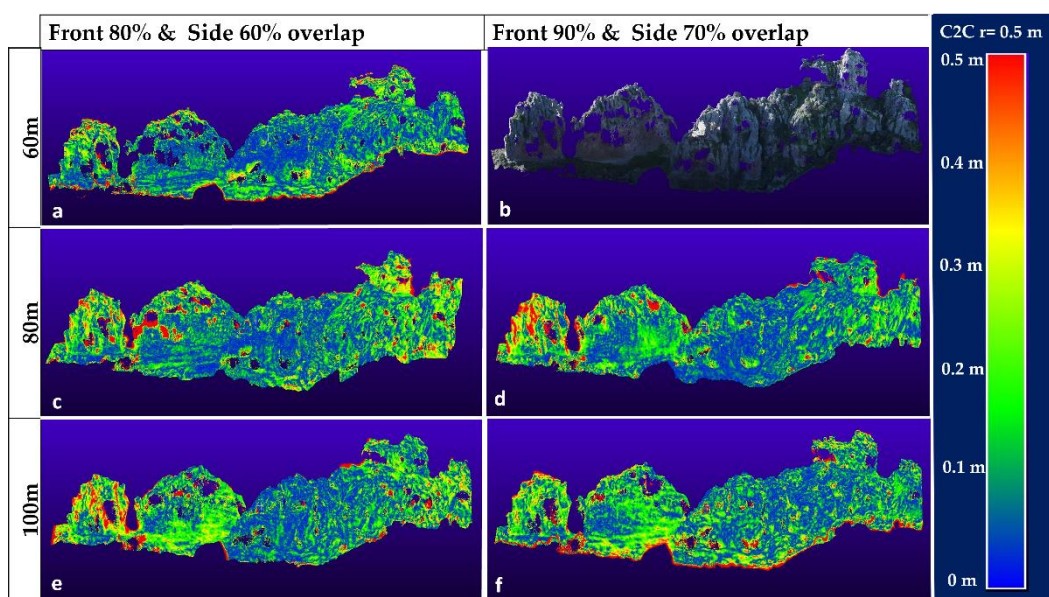

**Figure 10.** Results from the application of cloud-to-cloud absolute distance to 3D point clouds: (**a**) scenario 2 vs. scenario 1, (**b**) referenced point cloud, (**c**) scenario 2 vs. scenario 3, (**d**) scenario 2 vs. scenario 4, (**e**) scenario 2 vs. scenario 5, and (**f**) scenario 2 vs. scenario 6.

As in previous comparisons, when comparing the scenario 5 point cloud with the reference one, the center part of the gorge had no differences; thus, blue points having distances between 0 and 10 cm existed in this part. Red parts existed on the west side of the gorge, denoting high C2C distances (greater than 45 cm). Additionally, in the upper east part in both point clouds missing parts existed.

Finally, thed scenario 2 and scenario 6 comparison indicated similarity with the scenario 5 comparison results. Gaps apparent in the west part existed in the 3D dense point cloud of the gorge. In scenario 6, the green east parts denoted C2C distances that had values between 15 and 25 cm.

### 3.5. Geovisualization

The analysis of the 3D model also depended on the spatial resolution of the primary data. The 3D geometry of the digital object was formed by a triangulation irregular network to which the UAS acquired images then attributed a photorealistic texture. Each surface was lined with a part of the images that corresponded to the specific surface [59]. Therefore, the desired GSD resolution of 3 cm was crucial for the detailed geovisualization of the geosite at a scale of 1:400. Thus, the 3D model chosen for visualization was derived from Scenario 4 (altitude 80 m, 90% and 70% overlap) and its corresponding panorama.

Then, for the geovisualization of the geosite, the textured 3D model of the gorge and the panorama image were used. The 3D model was uploaded in the Sketchfab software [70] and acquired texture from the jpg file created from the high-resolution images collected with the UAS. One of the special features of the geosites' 3D models was that they represented outdoor areas. Therefore, the area of interest, when located in the outdoor space, was illuminated using as light sources (i) the sun, (ii) the sky, and (iii) the environmental lights [71]. The variety in landscape and the material reflectance created heterogeneous light sources that contributed to the light's total intensity that reflected and lightened the mapping area in the 3D space.

For this reason, the panorama of the area was chosen to act as environmental lights and was converted to a high dynamic range image (HDRI). A sphere produced from the

panorama had its center of gravity in the 3D model of the geosite. The HDRI takes the pixel's values and uses them as a light-emitting source, partially simulating a surrounding environment. Thus, the geosite 3D model was illuminated in the same way as it was illuminated at the time of the UAS data acquisition. Figure 11a shows a part of the 3D model that has been automatically visualized, where the shadows captured in the images were maintained in its texture. After using environmental lights in Figure 11b at the same part of the geosite's 3D model, the respective shadows appeared smoother and softer.

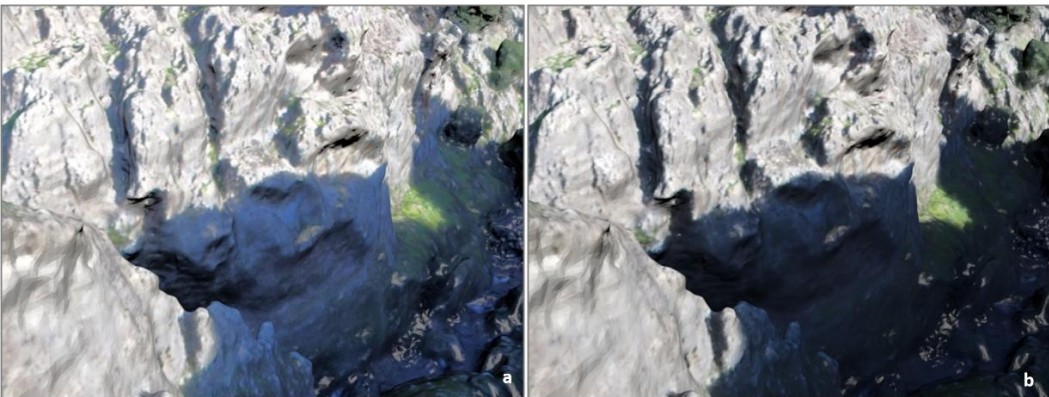

**Figure 11.** Part of the 3D model of Voulgaris gorge: (**a**) illuminated by environmental lights calculated by a panorama of the area, and (**b**) without lighting source.

For the geosite's visualization in a virtual environment, and for creating the augmented map, the Sketchfab software [70] was used. The scene created in the virtual space consisted of the following: (i) 3D model, (ii) texture, (iii) lights, and (iv) primary camera (https://skfb.ly/oqGYY accessed on 7 January 2021). The main camera defined the observer's position and the field of view from that exact position. We used the method of third-person viewer for the observer to illustrate the gorge's 3D model in the correct cartographic scale (1:400) on the augmented space. The observer–model distance was set to 80 m to achieve the best visualization of the geosite without unnecessary detail. Finally, an orthophoto map was used as a level on which to project the geosite's augmented 3D model (Figure 12). The AR gorge geovisualization presented in Figure 12 is accessible from any mobile phone as long as the SketchFab application and the 2D map of the study area are installed. The 3D model of the geosite is available to the public in the SketchFab repository and can be selected. Using the mobile phone's camera to point at the projection level (2D map) enhances the 3D model.

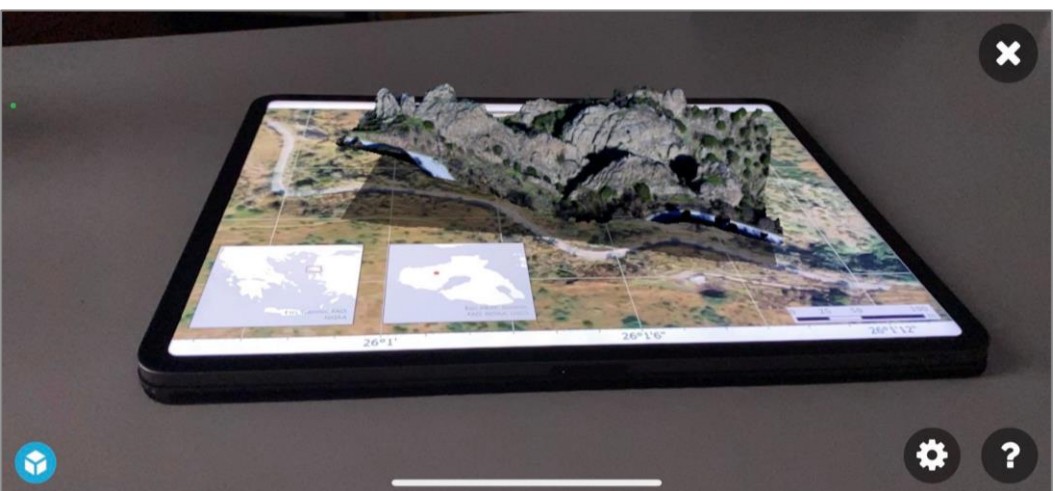

**Figure 12.** Augmented reality map of Voulgaris Gorge, geosite (scale 1:400).

## 4. Discussion

Creating 3D geovisualization of a geosite requires the proper acquisition of primary data images using a UAS. More specifically, for the 3D mapping of a geosite with sharp and irregular slopes such as the Voulgaris gorge, the above research shows that the coverage rate of 80% front and 60% side does not sufficiently cover the vertical sides of the area even when the flight altitude of UAS is low (60 m). None of the three flight altitudes, 60 m, 80 m, and 100 m, with this percentage of overlap managed to fully render the geological structure in the areas where negative slopes prevail. The images collected from the flights made at the respective altitudes (60 m, 80 m, and 100 m) with a coverage rate of 90% front lap and 70% side overlap produced 3D point clouds with a greater number of points on the steep surfaces, unsuitable for later creating the 3D geovisualization of the gorge. However, a significant absence of points was observed in the 3D point clouds created from the 60 m and 100 m flights. The collection of images for the creation of panoramas, although executed at three different altitudes, did not affect the results. The differences between the panoramas were minimal as the camera's lineup for capturing images was 360°. The use of panoramas as high dynamic range images was crucial for photorealistic geovisualization. It had been proven to help smooth out sharp and intense shadows and, more specifically, in areas with intense geomorphology and steep slopes where the shading effect intensifies. Transferring the 3D model of the geosite to a virtual reality environment was a challenge due to the gorge's geometry, making the user's information complex and confusing. Using the panorama as environmental lights for lighting the 3D model of the geosite helped in simulating the natural environment in the virtual space. When the 3D model was adequately illuminated, the virtual space became more user friendly and realistic.

The proposed methodology presented some limitations: (a) flight duration, (b) geomorphology of the study area, and (c) computer power. The total time of data collection lasted 2 h, resulting in capturing different shadows in the images. The shades then affected both the processing of the data and the texture rendered in the 3D models. Another difficulty encountered in this research was the geomorphology of the geosite, which made the study area approachable only from the south side. The available processing power capabilities significantly affected the data processing time, and the limitations imposed by the software used did not allow full utilization of the results. Nevertheless, the constant evolution of geovisualization means should have enabled full utilization of the 3D cartographic information.

## 5. Conclusions

Exploring scale issues prior to flight planning for data collection to create 3D geovisualizations in virtual space is essential in 3D mapping. More specifically, for collecting data/images for the 3D mapping of the geosite Voulgaris gorge, the optimal flight altitude was 80 m with 90% front lap and 70% side overlap for the 1:400 cartographic scale. Images collected from a higher altitude (100 m) lead to gaps appearing in the 3D point clouds, and shots from lower height (60 m) did not have enough front and side overlap to create complete 3D point clouds. Increased coverage percentage led to a more significant number of data collected, thus extending the processing time. Additionally, for creating the geosite's 3D model, the images' GSD should be $\frac{1}{4}$ of the spatial error (10 cm). In this survey, the images' GSD was 2.5 cm which was the most suitable for the 1:400 cartographic scale.

Also, the panorama's contribution was crucial for visualizing 3D outdoor models, especially in geosites with complex geomorphology such as the Voulgaris gorge. Environmental lights contributed to the uniform light distribution throughout the model, especially in steep and irregular slopes where shading was intense. The transfer of the properly illuminated 3D model with environmental lights calculated from a panorama of the area in an AR environment increased the photorealism of the gorge's geological structure and surrounding area.

The proposed UAS 3D geovisualization approach results were encouraging as the combination of scale-variant data acquisition flight plans and panoramas could offer an instrumental tool for realistic 3D mapping of geosites. To achieve the large-scale reproducibility of this framework, further research is needed to determine the critical limitations that influence data acquisition, such as sunlight conditions and the associated terrain shading effects.

Future goals might entail testing this specific methodology in areas with various types of geomorphology. It could also aid in exploring visual variables for the presentation of thematic information in the virtual space and creating contextual geovisualization for the promotion and management of geological sites.

**Author Contributions:** Conceptualization, E.-E.P., A.P., N.Z. and N.S.; methodology, E.-E.P. and N.S.; software, E.-E.P.; writing—original draft preparation, E.-E.P., A.P. and N.S.; writing—review and editing, all authors. All authors have read and agreed to the published version of the manuscript.

**Funding:** This research was funded by the Research e-Infrastructure "Interregional Digital Transformation for Culture and Tourism in Aegean Archipelagos" {Code Number MIS 5047046} which is implemented within the framework of the "Regional Excellence" Action of the Operational Program "Competitiveness, Entrepreneurship and Innovation". The action was co-funded by the European Regional Development Fund (ERDF) and the Greek State [Partnership Agreement 2014–2020].

**Acknowledgments:** We thank the editor and the three anonymous reviewers for their insightful comments which substantially improved the manuscript. We also thank Charalampos Psarros for his help with AR imaging and Aikaterini Rippi for her help with the English language editing.

**Conflicts of Interest:** The authors declare no conflict of interest.

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
