# Peer review of "Scale-Variant Flight Planning for the Creation of 3D Geovisualization and Augmented Reality Maps of Geosites: The Case of Voulgaris Gorge, Lesvos, Greece"

_applsci, doi:10.3390/app112210733_

Round 1

Reviewer 1 Report

389 The geosite 3D model is illuminated in the same way as it was illuminated at the time of the UAS data acquisition. Does this mean that the lighting and shadows on the images is from a different angle for each image since 2 hours of collecting data was needed? How that can be eliminated?  
455 The transfer of the properly illuminated 3D model with environmental lights calculated from a panorama of the area in an AR environment leads to a more enjoyable user experience compared to what other 3D geovizualization AR experiance?

Author Response

Dear reviewer,

Thank you for the effort to go through this paper and provide your useful feedback and valuable comments.
Following, you will find our replies to your comments.

Best regards,

Ermioni Eirini Papadopoulou

Reviewer 2 Report

Thank you for inviting me to evaluate the article titled “Scale Variant Flight Planning for the creation of 3D Geovisualization and Augmented Reality Maps of Geosites: The Case of Voulgaris Gorge, Lesvos, Greece”.

In my opinion, the research addresses an important and current issue, namely 3D geovisualization techniques. A new methods and new ways of usage of the geovisualization and its functionality are useful, and what is more very needed in pandemic days. The formal aspects of the paper are proper. The paper is well prepared and well-organized, it brings valuable results however, minor scientific revisions are needed.

The reviewed manuscript focuses on influence of cartographic scale and flight design in data acquisition using UAS to create augmented reality 3D geovisualization of geosites in Greece study area. The paper is well motivated and it might be relevant to the domain of applied earth sciences!

I suggest the authors revise the abstract. It would be worthwhile to expand the abstract with the results of the geovisualisation.

The introduction is well structured, and it covers all the concepts investigated in the methodological part.

I suggest the authors adding more details about the state of the art -- approaches and methods for geovisualization of geographic data, other project related to 3D visualization of data related to geovisualization would improve the paper.

Please indicate how your findings can be useful in other engineering disciplines, you can find some related paper in term of geovisualization here:

DOI: 10.3390/app11125466

DOI: 10.3390/app10196701

DOI: 10.3390/land10050492

The results and conclusions are correctly interpreted, and the discussions are logically related to the outcomes of the research aim. As mentioned before, I consider that this work brings added value in the field and the specific objectives of the manuscript are well related to the previous work developed in this domain.

My questions to the authors (not nessesary built into manuscript):

Can you explain what are the limitations of this method of data collecting (UAS) in field of geovisualization?

What processing power do you need to do 3D geovsualization?

Coming to other small observations:

  1. Figures 1, 2 and 10 have a poor quality. The text on these figures is very difficult to read.
  2. Table 2: instead of “&” use “and”
  3. Line 259: 3d=> 3D
  4.  In some part of Methodology (e.g. software links) is missing references
  5. I would ask the authors if the WebGIS is already explorable; in case it was, I would suggest adding the respective link to the paper (Figure 12).

My evaluation is that the paper is publishable after minor scientific revisions.

Author Response

Dear reviewer,

Thank you for taking the time to review our manuscript. We are grateful for the constructive feedback. Considering the comments, we have carefully made the revisions to incorporate your recommendations, and we elucidate more on the aspects that were not clear in the previous manuscript version.

Best regards,

Ermioni Eirini Papadopoulou

Reviewer 3 Report

Very good, interested, and original work. It Combine, UAV, AR, Digital Cartography, and photogrammetry. 

I like it reading it!

Some remarks:

  1. More explanations about the calculation of the spatial error of 20 cm are needed in my opinion.
  2. Thre is any correlation with the graphic error? (0,2 / 0,1 mm)
  3. How did you choose the flight speed? Has any impact on the quality of the results?
  4. There is a lot of software that has been used. You mentioned all. I would like to see a graph containing their use step by step. Finally where and how (using which hardware device?) can we see the result of the AR? We only see a photo...some more explanations are needed...
  5. Great job!

Author Response

Dear reviewer,

Thank you for taking the time to review our manuscript. We are grateful for the constructive feedback and the critical review, as well as for your valuable comments. In response, we carefully went through each one of your remarks and reworked the paper accordingly. Please find below our answer to your comments and the list of modifications that we have made.

Best regards,

Ermioni Eirini Papadopoulou
